# The Association between Early-Onset Diagnosis and Clinical Outcomes in Triple-Negative Breast Cancer: A Systematic Review and Meta-Analysis

**DOI:** 10.3390/cancers15071923

**Published:** 2023-03-23

**Authors:** Robert B. Basmadjian, Kristian Chow, Dayoung Kim, Matthew Kenney, Aysha Lukmanji, Dylan E. O’Sullivan, Yuan Xu, May Lynn Quan, Winson Y. Cheung, Sasha Lupichuk, Darren R. Brenner

**Affiliations:** 1Department of Community Health Sciences, Foothills Medical Centre, University of Calgary, Calgary, AB T2N 2T9, Canada; 2Department of Oncology, Tom Baker Cancer Centre, University of Calgary, Calgary, AB T2N 4N2, Canada; 3Department of Cancer Epidemiology and Prevention Research, Cancer Research & Analytics, Cancer Care Alberta, Alberta Health Services, Calgary, AB T2S 3C3, Canada; 4Department of Surgery, Foothills Medical Centre, University of Calgary, Calgary, AB T2N 2T9, Canada

**Keywords:** early-onset breast cancer, young age, triple-negative breast cancer, overall survival, breast cancer-specific survival, recurrence-free survival, pathological complete response, systematic review, meta-analysis

## Abstract

**Simple Summary:**

Epidemiologic studies have observed higher risks of breast cancer recurrence and death in women diagnosed <40 years of age compared to ≥40 years. There is ongoing clinical debate as to whether this is due to the overrepresentation of advanced and aggressive disease features or unique disease biology. Younger women are more likely to be diagnosed with more aggressive subtypes, including triple-negative breast cancer. Herein, we conduct a systematic review and meta-analysis of studies assessing the association between age <40 years at diagnosis and clinical outcomes in triple-negative breast cancer. We present pooled risk estimates for recurrence-free survival, breast cancer-specific and overall survival, and pathological complete response. Our findings highlight the prognostic significance of age in triple-negative breast cancer and may point to a need for tailored local and systemic treatment strategies in women of younger ages.

**Abstract:**

Early-onset diagnosis, defined by age <40 years, has historically been associated with inferior outcomes in breast cancer. Recent evidence suggests that this association is modified by molecular subtype. We performed a systematic review and meta-analysis of the literature to synthesize evidence on the association between early-onset diagnosis and clinical outcomes in triple-negative breast cancer (TNBC). Studies comparing the risk of clinical outcomes in non-metastatic TNBC between early-onset patients and later-onset patients (≥40 years) were queried in Medline and EMBASE from inception to February 2023. Separate meta-analyses were performed for breast cancer specific survival (BCSS), overall survival (OS), and disease-free survival (DFS), locoregional recurrence-free survival (LRRFS), distant recurrence-free survival (DRFS), and pathological complete response (pCR). In total, 7581 unique records were identified, and 36 studies satisfied inclusion criteria. The pooled risk of any recurrence was significantly greater in early-onset patients compared to later-onset patients. Better BCSS and OS were observed in early-onset patients relative to later-onset patients aged >60 years. The pooled odds of achieving pCR were significantly higher in early-onset patients. Future studies should evaluate the role of locoregional management of TNBC and the implementation of novel therapies such as PARP inhibitors in real-world settings, and whether they improve outcomes.

## 1. Introduction

Breast cancer is the most diagnosed malignancy and second most common cause of cancer death in Canadian women [1]. Approximately 6% of breast cancer diagnoses in Canada are early-onset, defined as a diagnosis before 40 years of age [1]. The risk of being diagnosed with breast cancer before age 40 is 0.6% (1 in 164), compared to a lifetime risk of 9.3% (1 in 11) [2]. Although diagnoses are uncommon in this age group, breast cancer is the most common cause of cancer death in women younger than 40 years [3,4,5].

Breast cancer arises from the accumulation of inherited and somatic mutations that result in uncontrolled cell division and proliferation. Over time, driver mutations, which are causally implicated in oncogenesis, confer selective advantages in early tumour cells and result in clonal expansion [6]. Driver mutations in over 30 cancer genes have been implicated in breast cancer, including *AKT1*, *BRCA1*, *CDH1*, *GATA3*, *PIK3C*, *PTEN*, *RB1*, and *TP53* [7,8]. Genetic and epigenetic changes also confer interactions between tumour cells and nearby tissue to facilitate tumour microenvironments that support tumour progression and metastasis [9]. Changes in the tumour microenvironment induce alterations in gene expression to adjust metabolic requirements of tumours for better adaptation. For example, mitochondrial activity is reduced to consume oxygen in hypoxic conditions and generate energy through oxidative phosphorylation [9,10].

Women with an early-onset diagnosis represent a patient population with high unmet clinical need. The evidence base for early-onset patients is limited given the demographics of the disease. While young women do participate in research studies, there are rarely enough young women in interventional settings to focus on this subset. Therefore, results to inform the treatment of young women are generally derived from findings among women of older age [11]. Despite being offered the same treatments, early-onset diagnosis has historically been associated with greater risks of local and contralateral recurrence, and worse survival compared to later-onset diagnosis after adjustment for important prognostic factors, including stage of diagnosis, tumour grade, histology, treatment, and surgery [12,13,14]. This has driven clinical debate as to whether breast cancer diagnosed at younger than 40 years represents a unique disease biology and requires more tailored treatment approaches [15].

Breast cancer is heterogenous in its prognosis and response to treatment. In 2000, a gene expression analysis by Perou et al. characterized breast tumour expression profiles that clustered into four distinct groups: luminal-like, normal-breast-like, human epidermal growth factor receptor 2 (HER2) enriched, and basal-like/triple-negative breast cancer (TNBC) [16]. Each subtype is associated with different prognoses, responses to treatment, and clinical outcomes. The advent of molecular subtyping has resulted in the development and validation of targeted therapies and improved outcomes, especially in luminal-like and HER2 subtypes. However, prognosis in TNBC is generally unfavourable [17]. Risk factors for TNBC include an earlier age of menarche, young age, higher body mass index in premenopausal years, higher parity, and lower duration of breast feeding [17,18]. Germline mutations in *BRCA1* and *BRCA2*, the most known breast cancer susceptibility genes, compromise DNA repair by homologous recombination and over 75% of tumours arising in women who inherit these mutations have a triple-negative phenotype [17,18]. TNBC is negative for estrogen receptors (ER), progesterone receptors (PR), and HER2, and cannot be treated with endocrine therapy or HER2-targeted therapy. Chemotherapy is the main systemic treatment available for TNBC [17]. However, the optimal regimes, specifically in young women in both the non-metastatic and metastatic settings, are currently unknown [11]. Many studies have demonstrated that a higher proportion of early-onset diagnoses are triple-negative compared to later-onset diagnoses, which may in part explain inferior outcomes among early-onset cases [12,13,14]. However, it is unknown whether early-onset diagnosis has significant prognostic implications within the TNBC subtype.

Recently, an increasing number of studies with large sample sizes and robust outcome data have assessed the prognostic significance of early-onset diagnosis by molecular subtype. For example, women <40 years with luminal B cancer have been consistently observed to have inferior outcomes compared to patients of older ages, while no age-related differences have been observed in the HER2 subtype since the introduction of targeted therapies [11,19,20]. Many studies have assessed the independent effects of a young age at diagnosis on outcomes in TNBC, but results in the current body of literature remain inconsistent and have yet to be systematically reviewed. Ascertaining the prognostic significance of young age in TNBC may have important implications for future research and clinical practice. It may provide insights as to whether TNBC in women <40 years is biologically unique, and whether tailored treatments are needed to improve outcomes in young women. We conducted a systematic review and meta-analysis to synthesize qualitative evidence of the association between early-onset diagnosis and clinical outcomes in non-metastatic TNBC.

## 2. Materials and Methods

The study was registered in PROSPERO (CRD42022308683). Reporting of this systematic review and meta-analysis adhered to the guidelines of Preferred Reporting Items for Systematic Reviews and Meta-Analyses (PRISMA) and Meta-analysis of Observational Studies in Epidemiology (MOOSE) [21,22]. Abstract screening and full-text review were completed independently in duplicate by three investigators (K.C., D.K., and M.K.). All conflicts were resolved by R.B.B. Quality assessment and data extraction were also completed independently in duplicate by three investigators (K.C., A.L., R.B.B.). All conflicts were solved by the 3 investigators via consensus.

### 2.1. Search Strategy

The Medline and Excerpta Medica Database (EMBASE) was queried from inception to 9 February 2023, the date of the last conducted search. We restricted our search to English language studies and no restrictions were made on publication date. The search strategy was developed by R.B.B., in consultation with librarians from the University of Calgary and Alberta Health Services. Search terms related to four key concepts: breast cancer, triple-negative subtype, clinical outcomes, and early-onset diagnosis. The search strategy is fully detailed in Appendix A.

### 2.2. Eligibility Criteria

Eligibility criteria were established in our PICOs (Population, Intervention, Comparator, and Outcomes) statement and prior to conducting the search. To be included, studies had to have compared clinical outcomes between early-onset and later-onset age groups in an adult (≥18 years) population of non-metastatic TNBC patients. An age of diagnosis of 40 years was used as a cut-off to define early-onset and later-onset age groups. Clinical outcomes included overall and breast cancer-specific survival; local, locoregional, and distant recurrences/events/metastases; and pathological complete response (pCR) following neoadjuvant chemotherapy. Prospective and retrospective cohort studies, case–control studies, cross-sectional studies, and pooled analyses of trial data were eligible for inclusion. In full-text screening, studies were excluded if they (1) were a published abstract, (2) did not include a non-metastatic TNBC population/subgroup, (3) defined early-onset patients by age >40 years, (4) did not compare clinical outcomes between early-onset and later-onset age groups in the TNBC group, and (5) did not report an estimate to quantify the association of early-onset age and outcomes or did not provide values necessary to derive an effect estimate.

### 2.3. Data Extraction

A data extraction form was created prior to conducting the search. Extracted information included author, year of publication, geographic location, data sources, study design, sample size, definitions of early-onset and late-onset age groups, outcomes examined, clinical stage, lymph node status, and receipt of surgery, radiotherapy, and neo/adjuvant chemotherapy. The referent later-onset age group, association estimates, and 95% confidence intervals (CIs) for early-onset age groups were also extracted. Extracted association estimates included hazard ratios (HR) and odds ratios (OR). Only adjusted estimates were extracted; however, unadjusted estimates were extracted if no adjusted estimates were available. All the extracted information was stored in an Excel file (Microsoft, Redmond, WA) and was checked for accuracy by R.B.B.

### 2.4. Quality Assessment

The quality of included studies was evaluated using the Newcastle Ottawa Scale (NOS) [23]. The NOS assesses three domains of internal validity: (1) selection of study groups (4 questions, 1 point each); (2) comparability of study groups (1 question, 2 points max); and (3) assessment of exposure and outcome (3 questions, 1 point each). A cumulative score of 0 indicates a study of low quality and a score of 9 indicates a study of high quality. For the assessment of comparability of study groups, a score of 0 was assigned for unadjusted estimates, a score of 1 was assigned for estimates adjusted for clinicopathological variables, and a score of 2 was assigned to estimates adjusted for clinicopathological and treatment variables.

### 2.5. Statistical Analysis

Random-effects models with DerSimonian and Laird weighting were employed to pool association estimates for all meta-analyses. Separate meta-analyses were conducted for each clinical outcome that was examined in 3 or more studies. If studies made comparisons with multiple early or later-onset age groups, then these estimates were pooled prior to meta-analysis. We identified 6 primary outcomes to obtain pooled estimates for (1) Disease-free survival (DFS)—time from diagnosis to first breast cancer related event/recurrence or death; (2) Locoregional recurrence-free survival (LRRFS)—time from diagnosis to local or locoregional recurrence; (3) Distant recurrence-free survival (DRFS)—time from diagnosis to distant recurrence or metastasis; (4) Breast cancer-specific survival (BCSS)—time from diagnosis to death due to breast cancer; (5) Overall survival (OS)—time from diagnosis to death due to any cause; (6) Pathological complete response (pCR)—the absence of invasive cancer in the breast or axilla.

Pooled hazard ratio (HR) estimates with 95% confidence intervals (95% CI) were obtained for DFS, LRRFS, DRFS, BCSS, and OS while pooled odds ratio (OR) estimates and 95% CIs were obtained for pCR. Individual study and pooled estimates were visualized using forest plots, and heterogeneity was quantified with the Q-test and I^2^ statistics. In anticipation of heterogeneity, we performed subgroup analyses among studies that defined later-onset age as >60 years if at least three estimates were available. It was hypothesized that later-onset age groups defined by >60 years would include a large portion of patients with comorbid conditions who received less intensive treatment or declined compared to later-onset age groups defined by ≥40 years. Therefore, to capture potential heterogenous effects, subgroup analysis was performed wherein one subgroup later-onset age was defined by age groups ≥40 years (<40 years vs. ≥40 years), and in the other subgroup later-onset age was defined by age groups >60 years (<40 years vs. >60 years). Meta-regressions, by definition of later-onset age group (>60 years vs. ≥40 years) and adjustment for confounding (scored 2 in the comparability domain of the NOS vs. 0 or 1), were performed to statistically test whether these variables contributed to between-study heterogeneity. A funnel plot was used to visually assess the potential of publication bias and was statistically tested using Egger’s regression. The “trim and fill” method was applied where significant publication was present. A *p* value of <0.05 was used to define significance for statistical tests. All analyses were performed using Stata version 16.

## 3. Results

The PRISMA flow diagram of the study selection process is depicted in Figure 1. The search strategy yielded 7581 unique records. Following abstract and full-text screening, 36 studies were included for meta-analysis. Studies were primarily excluded for the reasons of being a published/conference abstract, not including an early-onset TNBC population, and not comparing outcomes between early-onset and later-onset age groups in the TNBC population.

Table 1 presents characteristics of included studies [19,24,25,26,27,28,29,30,31,32,33,34,35,36,37,38,39,40,41,42,43,44,45,46,47,48,49,50,51,52,53,54,55,56,57,58]. Three studies were based on pooled data from clinical trials [32,43,51]. Early-onset age definitions included 20–29, 30–39, <35, ≤35, <40, and ≤40 years. Later-onset age groups included >35, ≥35, 35–50, ≥40, 40–49, 40–50, 40–60, >40, 41–50, 41–60, >50, ≥50, 50–59, 51–60, >60, >65, >74, and >75 years. Regarding clinical outcomes, sixteen studies assessed DFS, three studies assessed recurrence-free survival (RFS), one study assessed event-free survival (EFS), four studies assessed distant DFS (DDFS), two studies assessed DRFS, four studies assessed distant metastasis-free survival (DMFS), six studies assessed LRFS, three studies assessed LRRFS, one study assessed contralateral recurrence, eleven studies assessed BCSS, fifteen studies assessed OS, and six studies assessed pCR. Studies using real-world data scored well in the first domain of the NOS scale—the selection of study groups was representative of the underlying source population. Six studies reported unadjusted estimates, fourteen adjusted for clinicopathological variables only, and sixteen studies adjusted for treatment and clinicopathological variables.

Among studies that assessed DFS (Figure 2), there was an estimated 39% increase in the risk of any breast cancer event among early-onset patients compared to later-onset patients (HR = 1.39; 95% CI: 1.18–1.64; I^2^ = 53.9%, n = 20). This included studies assessing DFS, RFS, and EFS. There was also evidence that early-onset patients had significantly worse LRRFS (HR = 1.32; 95% CI: 1.01–1.72; I^2^ = 31.6%; n = 8) and DRFS (HR = 1.27; 95% CI = 1.04–1.56; I^2^ = 60.5%; n = 11) (Figure 3). The pooled HR estimate DRFS included studies assessing DDFS, DRFS, and DMFS. The meta-regression did not reveal that later-onset age and adjustment for confounding were significant sources of heterogeneity in meta-analyses of DFS, LRRFS, and DRFS (Appendix A–c). Publication bias was not observed in meta-analyses of DFS (*p* = 0.071), LRRFS (*p* = 0.116), and DRFS (*p* = 0.720) (Appendix A).

Meta-regression revealed that later-onset age contributed significant heterogeneity in meta-analyses of BCSS (*p* = 0.002) and OS (*p* = 0.024) (Appendix A). Therefore, subgroup analyses among studies where later-onset age was defined by >60 years and ≥40 years were conducted. Among studies that assessed BCSS with a later-onset age defined by >60, there was a 29% reduction in the risk of death due to breast cancer among early-onset patients (HR = 0.71; 95% CI: 0.54–0.94; I^2^ = 85.5%; n = 4). Conversely, the risk of death due to breast cancer was 12% higher among early-onset patients when later-onset age was defined by ≥40 years (HR = 1.12; 95% CI: 1.01–1.24; I^2^ = 36%; n = 10) (Figure 4). Similar results were observed among studies assessing OS, where the risk of all-cause mortality was 33% lower in the early-onset group when later-onset age was defined by age >60 years (HR = 0.67; 95% CI: 0.48–0.93; I^2^ = 79.2%; n = 3), but 17% higher when the later-onset age was defined by >40 years (HR = 1.17; 95% CI: 1.00–1.31; I^2^ = 66.5%; n = 14) (Figure 5). No evidence of publication bias was found for BCSS (*p* = 0.142) and OS (*p* = 0.174) (Appendix A).

Finally, the odds of achieving pCR were estimated to be 74% higher among early-onset patients compared to later-onset patients (OR = 1.74; 95% CI: 1.30–2.32; I^2^ = 75.2%; n = 6) (Figure 6). Evidence of publication bias was found for pCR (*p* = 0.004) (Appendix A). Following the trim and fill method, the adjusted pooled OR remained significant (OR = 1.46; 95% CI: 1.15–1.82).

## 4. Discussion

Early-onset diagnosis has historically been considered a poor prognostic factor in breast cancer. However, this association has been questioned with improved understanding of the biological heterogeneity of breast tumours, recognition of the predictive role of tumour subtype, and awareness that young women are more likely to develop more aggressive phenotypes. Our study expands on this body of evidence by systematically reviewing the literature on the association between the age of diagnosis and outcomes within the TNBC subtype. Overall, 36 studies were included in our meta-analyses, and we demonstrated that patients diagnosed with TNBC under 40 years of age are at greater risk of locoregional and distant recurrences compared to those diagnosed older than 40 years. We also show significantly better BCSS and OS among early-onset patients when compared to later-onset age groups defined by >60 years, but worse BCSS and OS when compared to later-onset age groups defined by ≥40 years. Finally, we found that early-onset patients are more likely to achieve pCR following neoadjuvant chemotherapy.

Given the aggressive features and poor prognosis of TNBC, there is concern that more aggressive treatment approaches should be considered. The absence of molecular markers for targeted therapy stresses the importance of locoregional management. Wang et al. performed a meta-analysis to evaluate the effectiveness of breast-conserving surgery (BCS) versus mastectomy in controlling locoregional recurrences for TNBC. Rates of locoregional recurrence and distant metastasis were 25% and 32% lower in the BCT group compared to the mastectomy group, respectively [59]. The benefit of BCS in TNBC may be attributable to the receipt of postoperative radiation. A meta-analysis by O’Rorke et al. demonstrated a significantly lower risk of locoregional recurrence in patients receiving adjuvant radiotherapy irrespective of surgery type versus mastectomy alone [60]. Improvements in locoregional recurrence were present in T1-2N0 and T3-4N2-3 subgroups, and survival benefits were observed in the <40 years subgroup [60]. Despite these findings, the role of postmastectomy radiation is not specific to subtype and often indicated for larger (T3) and node positive tumour, among other prognostic considerations. In the context of our results, the higher risk of locoregional and distant recurrence in early-onset patients may be driven by the fact that younger women are more likely to choose mastectomy [61,62] and may not be indicated for postmastectomy radiation for smaller, node negative disease. Cancello et al. describe T1N0 disease as a spectrum from patients at very low risk for whom there is little evidence supporting the use of adjuvant therapy, to those with higher risk disease where an approach including chemotherapy plus targeted therapy appears clearly justified [63]. In their study, both triple negativity and age <35 years were associated with worse LRRFS in T1N0 disease [63]. Studies included in our review did not report on the receipt of postmastectomy radiation, so it was not possible to ascertain whether this explained differences in recurrence rates between early and later-onset patients. Future studies should explore the role of postmastectomy radiation in young women with TNBC to further clarify optimal locoregional treatment approaches in this patient population.

High heterogeneity (>80%) was observed in pooled effect estimates for BCSS and OS, and was in part explained by varying age cut-offs to define later-onset diagnosis. In this review, we used an age cut-off of 40 years to define early-onset diagnosis, which is consistent with previous guidelines and reflects that these women have specific issues including those related to fertility, genetics and psychosocial concerns that often deserve a different approach compared to older premenopausal and postmenopausal women [11]. Most included studies defined early-onset age as <35 or <40 years. Conversely, the later-onset age groups varied more in definition, as some studies used an upper limit for the age range of the later-onset group and others did not. For example, studies in which the later-onset group was defined by an age range of 40–49 likely included higher proportions of premenopausal patients compared to later-onset groups defined as ≥40 years. Likewise, studies which used a high lower limit for defining the later-onset group, such as 60 or 75 years, included higher proportions who may have received less intensive (neo)adjuvant treatment or declined [64]. This likely explains why BCSS and OS were better among early-onset patients compared to later-onset patients >60 years in our subgroup analysis. Further, early-onset patients are more likely to undergo more aggressive management of recurrences. A multicentre French cohort including over 14,403 metastatic breast cancer patients, 28% of which were de novo diagnoses, reported a higher uptake of first-line chemotherapy in patients <40 years compared to >60 years (96.3% vs 90.2%) [65]. A study of patients with regional and distant recurrences in China observed similar findings, with an 88.7% uptake of first line chemotherapy in patients <40 years compared to 76% in those >65 years [66]. Early-onset patients are less likely to have pre-existing comorbid conditions or take multiple medications. Thus, they are generally healthier and can receive multiple rounds of combination chemotherapy and tolerate treatment-related toxicities that could result in non-compliance or declining treatment in others. Minor increases in the risk of breast cancer specific and all-cause mortality were observed in early-onset patients in subgroup analysis when compared to later-onset patients defined by ≥40 years. This may be due to the increased risk of distant recurrence/metastasis, which are considered not curable. TNBCs and young age have a higher propensity for metastases in the central nervous system, soft tissue organs, and multiple metastatic sites compared to bone alone, which is associated with better survival after metastasis [7,65,67,68,69]. If these occur more commonly in early-onset patients, poorer BCSS and OS would be expected. Previous studies have attributed this phenomenon to the “seed and soil” hypothesis, which states that tumor cells favour different microenvironments of distant organs, which provide ideal conditions for their invasion and proliferation [70,71]. The underlying molecular differences between early and later-onset TNBC patients should be explored in future studies to better understand the mechanisms of distant metastasis and whether these contribute to survival differences between these age groups.

Young women are underrepresented in clinical trials for novel therapies and outcome prediction tools to guide tailored treatment decisions [72]. Few actionable molecular targets exist for triple-negative disease, so systemic chemotherapy is generally recommended in addition to locoregional management. However, the optimal chemotherapy regime in young women in the early-stage setting is currently unknown [11]. Multicentre pooled analyses of early-stage patients treated with neoadjuvant chemotherapy demonstrate that young women with TNBC benefit most from preoperative treatment, given the prognostic value of achieving pCR [73,74]. Agreeable findings were observed in our meta-analysis of pCR. Recent data on the efficacy of adding platinum-based agents to neoadjuvant chemotherapy in early-stage TNBC has emerged from the BrighTNess trial, which showed improved EFS with the addition of carboplatin, with greater improvements in those <50 years of age versus ≥50 years [75]. Further, the KEYNOTE-522 trial showed that the addition of pembrolizumab, programmed death-1 (PD-1) targeted immunotherapy, to neoadjuvant chemotherapy improved EFS similarly in pre- and postmenopausal early-stage TNBC patients [76]. Of relevance to young women, who are more likely to harbour germline *BRCA1/2* mutations, are emerging data on the use of poly(ADP-ribose) polymerases (PARPs) inhibitors in early-stage breast cancer. The OLYMPIA trial randomized *BRCA1/2* mutation carriers to receive one year of adjuvant olaparib or placebo following completion of chemotherapy, surgery, and radiation [77]. The median age of the trial population was 43 years, and 82% of participants had triple-negative disease. Olaparib significantly improved invasive DFS (3-year rate, 85.9% vs. 77.1%; HR, 0.58; 99.5% CI, 0.41–0.82; *p* < 0.001) and OS (4-year rate, 89.8% vs. 86.4%; HR, 0.68; 98.5% CI, 0.47–0.97; *p* = 0.009) [77]. These trials demonstrate the discovery and implementation of novel therapies in TNBC with promising results in younger women; however, real-world studies will be needed in the future to assess whether they are generalizable to all early-onset TNBC patients. There is increasing attention in research and practice being paid to younger women with breast cancer, including a series of international consensus guidelines that encourage participation in clinical trials in the recurrent and advanced setting. Recent studies have observed higher participation for women <50 years in cancer clinical trials relative to 15–20 years ago [78,79]. Greater access to clinical trials and novel therapies may attenuate outcome differences in the future.

It is possible that tumors arising in young women have different biologic characteristics than do those that arise in older women, even within tumor subtypes. TNBCs are a highly diverse group of cancer, with differential responses to treatment [80]. Lehmann et al. were among the first to identify distinct subtypes of TNBC and their implications on selection for neo/adjuvant treatment [81]. Their comprehensive analyses demonstrated four subtypes of TNBC with unique gene expression profiles—basal-like 1 and 2 (BL1 and BL2), mesenchymal (M), and luminal androgen receptor (LAR) type. Non-basal TNBC tumors were diagnosed in older women relative to basal TNBC (58.5 vs. 52.6, *p* < 0.0001) [81]. The LAR subtype was diagnosed in women of older ages compared to all other subtypes (59.5 vs. 52.7, *p* < 0.0001) [81]. Concordant findings were observed by Prat et al., who showed that TNBCs can be grouped into claudin-low (high expression of mesenchymal processes), basal-like, and luminal/HER2 enriched subgroups [82]. In their data, the mean age at diagnosis of basal-like versus non-basal-like TNBC was found to be significantly different (50.7 vs. 57.1 years; *p* < 0.0001) [82]. Gulbache et al. compared the distribution of TNBC subtypes by age groups of <50, 50–64, and ≥65 years. A significantly higher proportion of basal-like TNBCs were observed in the <50 year age group, while a higher proportion of luminal and HER2 enriched subtypes were observed in the 50–64 and ≥65 year age groups [83]. They also used qRT-PCR and evaluated a set of 10 proliferation-related genes: *ANLN*, *CDC20*, *CENPF*, *CEP55*, *KIF2C*, *RRM2*, *UBE2C*, *MKI67*, *CCNB1*, and *MYBL2*. These proliferation genes correlate with Ki67 expression, and the last four genes are also included in the set of genes of commonly used and commercially available Oncotype DX testing. All 10 proliferation genes had significantly lower expression among older women with TNBC [83]. These results are consistent with studies using IHC staining, which report that older women with TNBC had a higher frequency of cytokeratin 5/6 expression, lower expression of EGFR, a lower rate of Ki67 labeling index and cytokeratin 7/8 positivity, and a higher rate of Bcl2 and cytokeratin 18 positivity [83,84,85]. A higher proportion of the young TNBC patients harbor a *BRCA1/2* mutation; however, its role is debatable and a larger prospective trial showed that survival after two years is more favorable among BRCA mutation carriers with TNBC compared with wild-type TNBC, but not at five years [48,86].

A major strength of this review is that all included studies were published in 2010 or later and likely reflect the administration of modern-day treatment regimens. Another strength is that most included studies scored moderate to high in our quality assessment. The majority of studies included representative study samples, mutually adjusted effect estimates, and the accurate collection of exposure and outcome data. Despite their high quality, the observational nature of these studies makes them prone to unmeasured confounding and biases. For example, unmeasured confounding is likely present in studies that did not account for treatment and dose of treatments. Another limitation includes the varying definitions of older-onset age groups among included studies, increasing the heterogeneity of pooled estimates, affecting the validity of their clinical interpretations.

## 5. Conclusions

Our systematic review provides evidence that patients diagnosed <40 years have worse LRRFS, DDFS, BCSS and OS compared to patients diagnosed ≥40 years in TNBC. However, when compared to later-onset patients >60 years, BCSS and OS are significantly better in early-onset patients. We also show greater odds of achieving a pCR in early-onset patients. While we speculate that differences in recurrence risk may be due to age-related patient preferences for mastectomy and physician bias in radiotherapy recommendation, further prospective data are needed to tailor locoregional treatments by age and molecular subtype. The tolerability of early-onset patients for more aggressive management, particularly in recurrent settings, may explain better survival outcomes compared to patients >60 years. Survival in TNBC remains poor irrespective of age, and the discovery of molecular markers has led the development of novel targeted therapies which must be further evaluated in real-world practice to determine if they are improving outcomes. Age-related differences in the biology of TNBC are being discovered, but their clinical relevance in terms of pathology and targets for treatment remains to be seen.

## Figures and Tables

**Figure 1 cancers-15-01923-f001:**
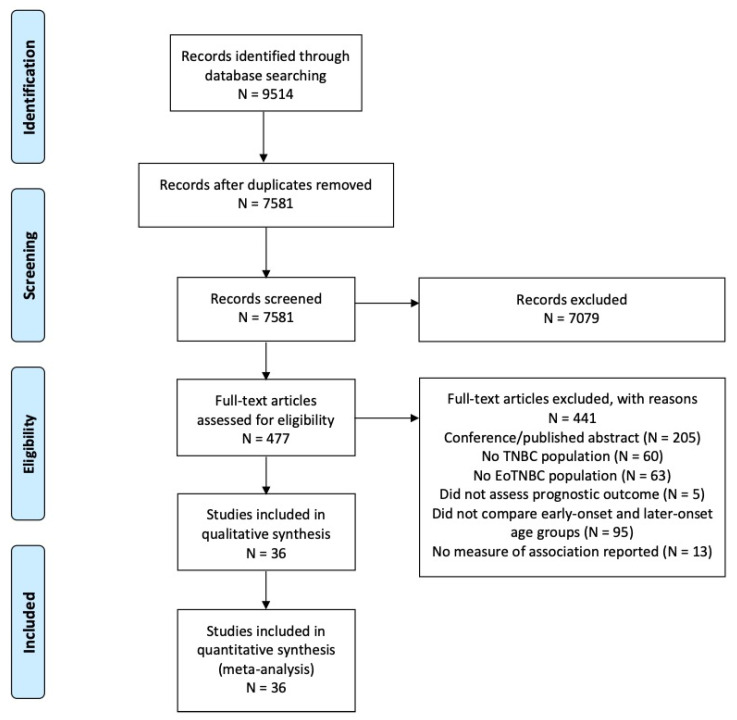
PRISMA flow diagram of included and excluded studies based on eligibility criteria [22].

**Figure 2 cancers-15-01923-f002:**
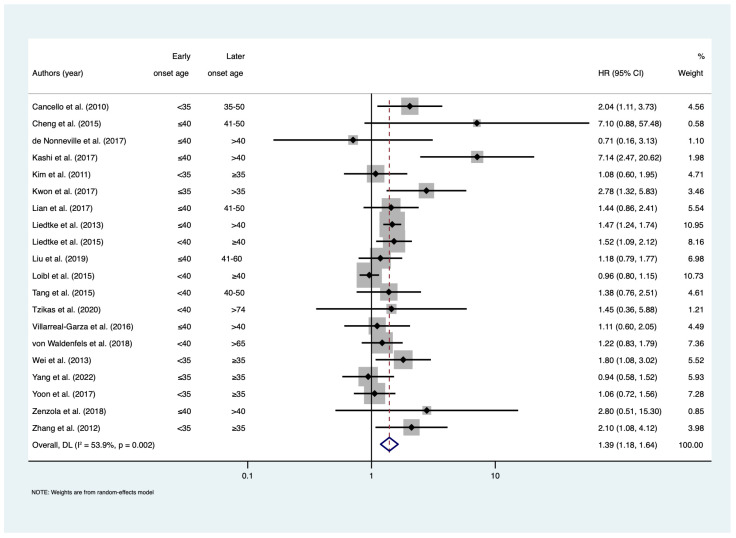
Forest plot for meta-analysis of disease-free survival. Hazard ratios less than 1 show favourable outcomes in early-onset age group; hazard ratios greater than 1 show poorer outcomes in early-onset age group. References [20,21,23,28,29,31,33,34,35,36,37,41,42,44,45,46,47,49,51,52]. Abbreviations: CI = confidence interval; HR = hazard ratio.

**Figure 3 cancers-15-01923-f003:**
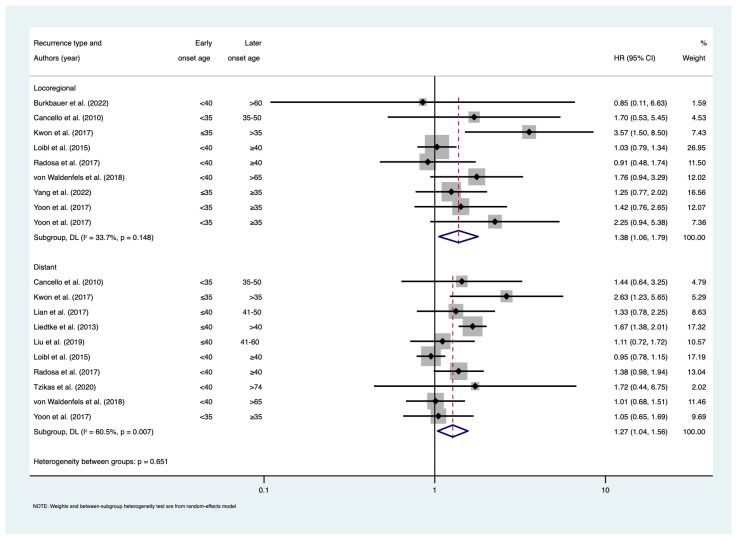
Forest plot for meta-analysis of locoregional and distant recurrence-free survival. Hazard ratios less than 1 show favourable outcomes in early-onset age group; hazard ratios greater than 1 show poorer outcomes in early-onset age group. References [19,20,31,33,34,36,37,38,42,45,47,49]. Abbreviations: CI = confidence interval; HR = hazard ratio.

**Figure 4 cancers-15-01923-f004:**
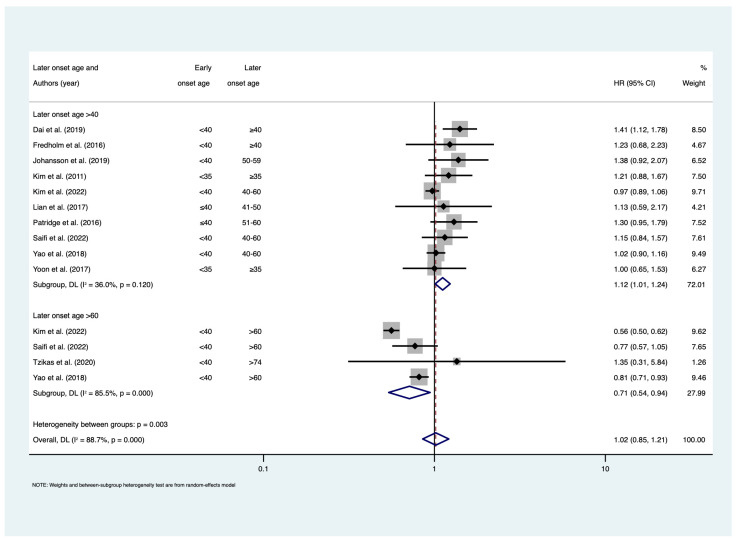
Forest plot for meta-analysis of breast cancer-specific survival. Hazard ratios less than 1 show favourable outcomes in early-onset age group; hazard ratios greater than 1 show poorer outcomes in early-onset age group. References [13,22,25,27,29,30,33,40,42,48,49]. Abbreviations: CI = confidence interval; HR = hazard ratio.

**Figure 5 cancers-15-01923-f005:**
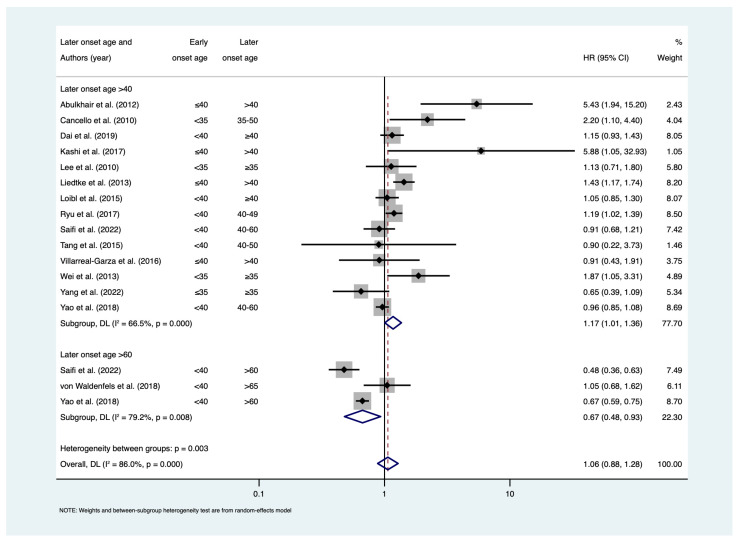
Forest plot for meta-analysis of overall survival. Hazard ratios less than 1 show favourable outcomes in early-onset age group; hazard ratios greater than 1 show poorer outcomes in early-onset age group. References [18,20,22,28,32,34,37,39,40,41,44,45,46,47,48]. Abbreviations: CI = confidence interval; HR = hazard ratio.

**Figure 6 cancers-15-01923-f006:**
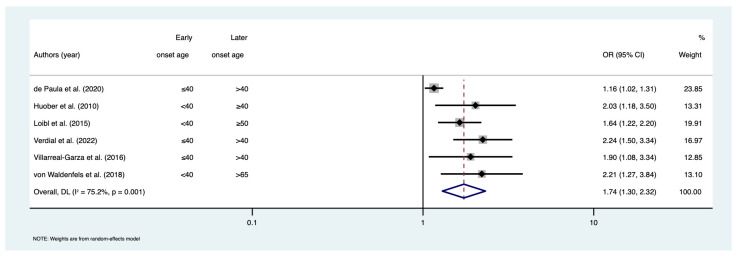
Forest plot for meta-analysis of pathological complete response (pCR). Odds ratios less than 1 show lower odds of achieving pCR in early-onset age group; Odds ratios greater than 1 show higher odds of achieving pCR in early-onset age group. References [24,26,37,43,44,45]. Abbreviations: CI = confidence interval; OR = odds ratio.

**Table 1 cancers-15-01923-t001:** Characteristics and quality assessment of included studies.

Authors	Publication Year	Study Start–End	Location	TNBC Size	EoTNBC Age Group(s)	LoTNBC Age Group(s)	Outcome Type(s)	Effect Estimate (95% CI)	Newcastle–Ottawa Scale Assessments							
									Selection				Comparability	Outcome		
									Q1	Q2	Q3	Q4	Q1	Q1	Q2	Q3
Abulkhair et al. [24]	2012	2001–2008	Saudi Arabia	26	≤40	>40	OS	5.43 (1.94–15.2)	1	1	1	1	1	1	1	0
Burkbauer et al. [25]	2022	2009–2018	United States	648	<40	>60	LRFS	0.85 (0.11–6.69)	1	1	1	1	1	1	1	1
Cancello et al. [26]	2010	1997–2004	Italy	251	<35	35–50	OS	2.2 (1.1–4.41)	1	1	1	1	2	1	1	0
							DFS	2.04 (1.11–3.72)								
							LRRFS	1.7 (0.53–5.44)								
							DMFS	1.44 (0.64–3.27)								
Cheng et al. [27]	2015	1990–2008	Taiwan	171	≤40	41–50	RFS	7.1 (0.90–59)	1	1	1	1	0	1	1	0
Dai et al. [28]	2019	2010–NR	China	378	<40	≥40	OS	1.15 (0.93–1.43)	1	1	1	1	2	1	1	0
							BCSS	1.41 (1.12–1.79)								
de Nonneville et al. [29]	2017	1987–2013	France	284	≤40	>40	DFS	0.71 (0.16–3.10)	1	1	1	1	2	1	1	0
de Paula et al. [30]	2020	2010–2013	Brazil	NR	≤40	>40	pCR	1.16 (1.02–1.31)	1	1	1	1	1	1	1	0
Fredholm et al. [31]	2016	1992–2005	Sweden	152	<40	≥40	BCSS	1.23 (0.68–2.23)	1	1	1	1	0	1	1	0
Huober et al. [32]	2010	2002–2005	Europe	378	<40	≥40	pCR	2.03 (1.18–3.5)	0	1	1	1	2	1	1	1
Johansson et al. [33]	2019	2005–2015	Norway	2030	<40	50–59	BCSS	1.38 (0.92–2.07)	1	1	1	1	1	1	1	1
Kashi et al. [34]	2017	2002–2014	Iran	180	≤40	>40	DFS	7.14 (4.00–33.33)	1	1	1	1	1	1	1	0
							OS	5.88 (1.06–33.33)								
Kim et al. [35]	2011	2000–2005	Korea	513 5806	<35	≥35	RFS	1.08 (0.6–1.95)	1	1	1	1	2	1	1	0
		1993–2006			<35	≥35	BCSS	1.21 (0.88–1.67)								
Kim et al. [36]	2022	2010–2015	United States	29,893	<40	40–60	BCSS	0.97 (0.88–1.06)	1	1	1	1	2	1	1	1
						61–75	BCSS	0.55 (0.49–0.62)								
						>75	BCSS	0.58 (0.47–0.71)								
Kwon et al. [37]	2017	2003–2012	Korea	233	≤35	>35	DFS	2.78 (1.33–5.88)	1	1	1	1	1	1	1	0
							LRFS	3.57 (1.47–8.33)								
							DMFS	2.63 (1.20–5.56)								
Lee et al. [38]	2010	1993–2008	Korea	5586	<35	35–50	OS	1.03 (0.54–1.96)	1	1	1	1	1	1	1	0
						>50	OS	1.26 (0.65–2.45)								
Lian et al. [39]	2017	2004–2007	China	82	≤40	41–50	DFS	1.44 (0.86–2.40)	1	1	1	1	2	1	1	0
							DRFS	1.33 (0.78–2.24)								
							BCSS	1.13 (0.59–2.17)								
Liedtke et al. [40]	2013	1982–2008	United States	1732	≤40	>40	OS	1.43 (1.16–1.72)	1	1	1	1	2	1	1	0
							DDFS	1.67 (1.37–2.00)								
							DFS	1.47 (1.23–1.72)								
Liedtke et al. [41]	2015	Varying Periods	International	783	<40	≥40	EFS	1.52 (1.09–2.12)	1	1	1	1	0	1	1	1
Liu et al. [42]	2019	2000–2016	United States	94	≤40	41–60	DFS	1.18 (0.79–1.78)	1	1	1	1	2	1	1	0
							DMFS	1.11 (0.72–1.72)								
Loibl et al. [43]	2015	1998–2010	Germany	1645	<40	≥50	pCR	1.64 (1.22–2.19)	0	1	1	1	2	1	1	1
						40–49	DDFS	0.87 (0.66–1.15)								
						≥50	DDFS	1.03 (0.79–1.35)								
						40–49	LRFS	1.03 (0.70–1.52)								
						≥50	LRFS	1.03 (0.71–1.49)								
						40–49	DFS	0.93 (0.72–1.20)								
						≥50	DFS	0.99 (0.78–1.28)								
						40–49	OS	0.97 (0.72–1.32)								
						≥50	OS	1.14 (0.85–1.52)								
Patridge et al. [19]	2016	2000–2007	United States	2886	≤40	51–60	BCSS	1.3 (0.90–1.70)	1	1	1	1	2	1	1	1
Radosa et al. [44]	2017	1998–2011	United States	1930	<40	≥40	LRFS	0.91 (0.41–1.75)	1	1	1	1	2	1	1	0
							DRFS	1.38 (0.99–1.95)								
Ryu et al. [45]	2017	2003–2010	Korea	5875	20–29	40–49	OS	1.14 (0.76–1.71)	1	1	1	1	2	1	1	1
					30–39	40–49	OS	1.20 (1.01–1.42)								
Saifi et al. [46]	2022	2010–2015	United States	12,761	<40	40–60	BCSS	1.15 (0.84–1.56)	1	1	1	1	1	1	1	1
						>60	BCSS	0.77 (0.59–1.09)								
						40–60	OS	0.91 (0.71–1.27)								
						>60	OS	0.48 (0.35–0.62)								
Tang et al. [47]	2015	2003–2012	China	672	<40	40–50	OS	0.9 (0.22–3.77)	1	1	1	1	1	1	1	0
							DFS	1.38 (0.76–2.52)								
Tzikas et al. [48]	2020	2007–2015	Sweden	524	<40	>74	RFS	1.45 (0.36–5.88)	1	1	1	1	2	1	1	1
							DDFS	1.72 (0.43–6.67)								
							BCSS	1.35 (0.31–5.88)								
Verdial et al. [49]	2022	2013–2018	United States	394	≤40	41–60	pCR	2.01 (1.21–3.34)	1	1	1	1	0	1	1	1
						≥61	pCR	2.63(1.4–5.09)								
Villarreal-Garza et al. [50]	2016	2007–2013	Mexico	287	≤40	>40	pCR	1.9 (1.1–3.4)	1	1	1	1	1	1	1	0
							OS	0.91 (0.45–2.00)								
							DFS	1.11 (0.59–2.00)								
von Waldenfels et al. [51]	2018	1998–2010	Germany	1638	<40	>65	DDFS	1.01 (0.68–1.51)	0	1	1	1	2	1	1	1
							LRRFS	1.76 (0.94–3.29)								
							DFS	1.22 (0.83–1.78)								
							OS	1.05 (0.68–1.61)								
							pCR	2.21 (1.27–3.84)								
Wei et al. [52]	2013	2002–2004	China	309	<35	≥35	DFS	1.80 (1.08–3.02)	1	1	1	1	0	1	1	0
							OS	1.87 (1.05–3.31)								
Yang et al. [53]	2022	2008–2018	China	1158	<35	35–50	LRFS	1.49 (0.75–2.49)	1	1	1	1	0	1	1	1
						>50	LRFS	1.05 (0.54–2.08)								
						35–50	DFS	1.20 (0.77–1.89)								
						>50	DFS	0.74 (0.47–1.14)								
						35–50	OS	0.85 (0.46–1.59)								
						>50	OS	0.51 (0.28–0.93)								
Yao et al. [54]	2018	2010–2014	China	22,802	<40	40–60	BCSS	1.02 (0.89–1.15)	1	1	1	1	2	1	1	0
						>60	BCSS	0.81 (0.71–0.93)								
						40–60	OS	0.96 (0.85–1.09)								
						>60	OS	0.67 (0.59–0.75)								
Yoon et al. [55]	2017	1989–2008	Korea	1792	<35	≥35	DFS	1.06 (0.72–1.56)	1	1	1	1	1	1	1	0
							LRRFS	1.42 (0.76–2.65)								
							LRFS	2.25 (0.94–5.38)								
							DMFS	1.05 (0.65–1.68)								
							BCSS	1.00 (0.65–1.53)								
Yoon et al. [56]	2019	1989–2008	Korea	845	<35	≥35	Contralateral breast recurrence	2.00 (1.20–3.33)	1	1	1	1	1	1	1	0
Zenzola et al. [57]	2018	1999–2014	Spain	201	≤40	>40	DFS	2.80 (0.51–15.31)	1	1	1	1	1	1	1	0
Zhang et al. [58]	2012	2003–2004	China	356	<35	≥35	DFS	2.11 (1.08–4.13)	1	1	1	1	1	1	1	0

Effect estimates for all outcomes were hazard ratios, except pathological complete response which were odds ratios. Abbreviations: BCSS = breast cancer-specific survival; DFS = disease-free survival; DDFS = distant disease-free survival; DMFS = distant metastasis- free survival; DRFS = distant recurrence-free survival; EFS = event-free survival; Eo = early-onset; Lo = later-onset; LRFS = local recurrence-free survival; LRRFS = locoregional recurrence-free survival; NR = not reported; OS = overall survival; pCR = pathological complete response; RFS = recurrence-free survival; TNBC = triple negative breast cancer.

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
