# Peer review of "The Association between Early-Onset Diagnosis and Clinical Outcomes in Triple-Negative Breast Cancer: A Systematic Review and Meta-Analysis"

_cancers, 2023, doi:10.3390/cancers15071923_

Round 1

Reviewer 1 Report

The authors present a systematic review and meta-analysis of the association between early-onset diagnosis and clinical outcomes in triple-negative breast cancer.

They investigate in their well-written and concise manuscript the association between age < 40 years at diagnosis and clinical outcomes in triple-negative breast cancer using pooled risk estimates for recurrence-free survival, breast cancer-specific and overall survival, and pathological complete response. They are able to show that the pooled odds of achieving a pathologic complete response and the pooled risk of any recurrence was significantly greater in early-onset patients compared to later-onset patients (≥40 years). Conversely, breast cancer-specific survival and overall survival improved in patients with early onset compared with patients with late onset older than 60 years.

The systematic review and meta-analysis are well conducted and the results are clearly and thouroughly discussed. The results are important, although expected. However, I have a single comment:

1)    It is confusing to the reader that the authors use two different definitions for patients with later onset (≥40 years and >60 years). Even if they find significant heterogeneity, the message would be clearer if they stuck to a single cutoff between early and late onset.

The authors present a systematic review and meta-analysis of the association between early-onset diagnosis and clinical outcomes in triple-negative breast cancer.

They investigate in their well-written and concise manuscript the association between age < 40 years at diagnosis and clinical outcomes in triple-negative breast cancer using pooled risk estimates for recurrence-free survival, breast cancer-specific and overall survival, and pathological complete response. They are able to show that the pooled odds of achieving a pathologic complete response and the pooled risk of any recurrence was significantly greater in early-onset patients compared to later-onset patients (≥40 years). Conversely, breast cancer-specific survival and overall survival improved in patients with early onset compared with patients with late onset older than 60 years.

The systematic review and meta-analysis are well conducted and the results are clearly and thouroughly discussed. The results are important, although expected. However, I have a single comment:

1)    It is confusing to the reader that the authors use two different definitions for patients with later onset (≥40 years and >60 years). Even if they find significant heterogeneity, the message would be clearer if they stuck to a single cutoff between early and late onset.

The authors present a systematic review and meta-analysis of the association between early-onset diagnosis and clinical outcomes in triple-negative breast cancer.

They investigate in their well-written and concise manuscript the association between age < 40 years at diagnosis and clinical outcomes in triple-negative breast cancer using pooled risk estimates for recurrence-free survival, breast cancer-specific and overall survival, and pathological complete response. They are able to show that the pooled odds of achieving a pathologic complete response and the pooled risk of any recurrence was significantly greater in early-onset patients compared to later-onset patients (≥40 years). Conversely, breast cancer-specific survival and overall survival improved in patients with early onset compared with patients with late onset older than 60 years.

The systematic review and meta-analysis are well conducted and the results are clearly and thouroughly discussed. The results are important, although expected. However, I have a single comment:

1)    It is confusing to the reader that the authors use two different definitions for patients with later onset (≥40 years and >60 years). Even if they find significant heterogeneity, the message would be clearer if they stuck to a single cutoff between early and late onset. The authors should make clearer, why theyused two different cut-offs

v

Author Response

Thank you for this comment. The primary definition of later-onset age in this study is ≥40 years. However, we perform subgroup analysis where early-onset age is compared to later-onset age defined by ≥40 years (<40 vs. ≥40 years), and where early-onset age is compared to later-onset age defined by >60 years (<40 vs. >60 years), if at least three estimates were available. We used two cut-offs for clinical and methodological reasons. From a clinical perspective, later-onset age groups defined >60 years are likely to include a large portion of patients who received less intensive treatment or declined and would be systematically different from patients in later-onset groups defined by ≥40 years. To this point, it was hypothesized that heterogeneity would arise from study estimates where <40 years were compared to >60 years and study estimates where <40 years were compared to ≥40 years. To provide more clarity for the rationale of using two cut-offs, the following statements were added to the Statistical Analysis section of the Methods:

“In anticipation of heterogeneity, we performed subgroup analysis among studies that defined later-onset age as >60 years if at least three estimates were available. It was hypothesized that later-onset age groups defined by >60 years would include a large portion of patients with comorbid conditions and received less intensive treatment or declined compared to later-onset age groups defined by ≥40 years. Therefore, to capture potential heterogenous effects, subgroup analysis was performed where in one subgroup later-onset age was defined by age groups ≥40 years (<40 years vs. ≥40 years), and in the other subgroup later-onset age was defined by age groups >60 years (<40 years vs. >60 years).”

Reviewer 2 Report

Dear Authors:

The manuscript "The association between early-onset diagnosis and clinical outcomes in triple-negative breast cancer: a systematic review and meta-analysis" by Basmadjian et al has demonstrated that prognostic significance of age in triple-negative breast cancer and may point to a need for tailored local and systemic treatment strategies in women of younger ages. I have just a few suggestions.

In introduction, please add more background information about breast cancer (Please cite:

1. An Overview: The Diversified Role of Mitochondria in Cancer Metabolism. Int J Biol Sci. 2023 Jan 16;19(3):897-915. doi: 10.7150/ijbs.81609. PMID: 36778129; PMCID: PMC9910000.

2. Advances in the Prevention and Treatment of Obesity-Driven Effects in Breast Cancers. Front Oncol. 2022 Jun 22;12:820968. doi: 10.3389/fonc.2022.820968. PMID: 35814391; PMCID: PMC9258420.

3. Mitochondrial mutations and mitoepigenetics: Focus on regulation of oxidative stress-induced responses in breast cancers. Semin Cancer Biol. 2022 Aug;83:556-569. doi: 10.1016/j.semcancer.2020.09.012. Epub 2020 Oct 6. Erratum in: Semin Cancer Biol. 2022 Nov;86(Pt 2):1222. PMID: 33035656.)

Best,

Author Response

Thank you for the suggestions. Several statements have been added to the Introduction with these recommended citations:

“Breast cancer arises from the accumulation of inherited and somatic mutations that result in uncontrolled cell division and proliferation. Over time, driver mutations, which are causally implicated in oncogenesis, confer selective advantages in early tumour cells and result in clonal expansion.[6] Driver mutations in over 30 cancer genes have been implicated in breast cancer, including AKT1, BRCA1, CDH1, GATA3, PIK3C, PTEN, RB1, and TP53.[7, 8] Genetic and epigenetic changes also confer interaction between tumour cells and nearby tissue to facilitate tumour microenvironments that support tumour progression and metastasis.[9] Changes in the tumour microenvironment induce alterations in gene expression to adjust metabolic requirements of tumours for better adaptation. For example, mitochondrial activity is reduced to consume oxygen in hypoxic conditions and generate energy through oxidative phosphorylation.[9, 10]”

“Risk factors for TNBC include earlier age of menarche, young age, higher body mass index in premenopausal years, higher parity, and lower duration of breast feeding.[17, 18] Germline mutations in BRCA1 and BRCA2, the most known breast-cancer susceptibility genes, compromise DNA repair by homologous recombination and over 75% of tumours arising in women who inherit these mutations have a triple-negative phenotype.[17, 18]”

Round 2

Reviewer 2 Report

Strongly suggest for publication.